# Fistula in War-Torn Tigray: A Call to Action

**DOI:** 10.3390/ijerph192315954

**Published:** 2022-11-30

**Authors:** Hailay Abrha Gesesew, Kenfe Tesfay Berhe, Simon Gebretsadik, Melaku Abreha, Mebrahtom Haftu

**Affiliations:** 1Research Centre for Public Health, Equity and Human Flourishing, Torrens University Australia, Adelaide, SA 5000, Australia; 2College of Health Sciences, Mekelle University, Mekelle 231, Tigray, Ethiopia; 3Gender Based Violence Team, Tigray Regional Health Bureau, Mekelle 07, Tigray, Ethiopia; 4Hamlin Fistula Centre, Mekelle 3609, Tigray, Ethiopia

**Keywords:** fistula, Tigray war, conflict, Hamlin Fistula Centre

## Abstract

Fistula is roaring in the ongoing war on Tigray. The potential risk factors for fistula in the conflict zone include obstructed labour due to limited or absent maternal care services, a correlation between malnutrition-stunted growth and birth difficulties and trauma, and sexually transmitted infections (STIs) due to conflict-related sexual violence. As a call to action to mitigate the unimaginable suffering that women and girls are facing in the region, concerted international effort is needed to provide treatment, rehabilitation, and re-integration; secure peace and stability; rebuild the health-care system; and ensure perpetrators are held accountable.

## 1. Background

The devastating war on Tigray, North Ethiopia, that started in early November 2020 has resulted in unimaginable atrocities and immeasurable consequences for the region’s humans, animals, and ecosystems [1,2]. Focusing on the impacts on human beings, thousands of civilians have died due to direct killings and lack of medicine and food [1,3]; the incidences and prevalence of morbidities from existing and new illnesses are increasing [4]; and people are suffering due to the near-total collapse of the health-care system [4,5]. Coupled with the imposed siege of the entire region by the Ethiopian and Eritrean governments and Amhara regional forces, women and children living amongst the ongoing conflict in war-torn Tigray are the ones paying the highest price [6]. In particular, women who have developed fistula experience excruciating pain as a result of isolation, shame, and rejection by families and communities [7], partly due to fecal/urinary incontinence.

A vaginal fistula is an unusual opening that connects the vagina to another organ, such as the bladder (vesicovaginal fistula), ureters (ureterovaginal fistula), urethra (urethrovaginal fistula), rectum (rectovaginal fistula), large intestine (colovaginal fistula), or small intestine (enterovaginal fistula) [8,9]. There are several risk factors that determine fistula and, sadly, the women living in war-torn Tigray are susceptible to almost all of them [7,8]. Some public health facilities and non-governmental organizations (NGOs) in the region have included fistula in their community mobilization programs. A rare successful satellite-phone conversation with a clinician working in Mekelle’s Hamlin Fistula Centre, a center established in 2006 in Mekelle dedicated to treating and caring for women with fistula, revealed that about 250 war-victim women and girls had been admitted to the clinic (Personal communication, 2022). The clinician added, “… there are many women who are suffering at home. We are for example mobilizing community to report victims and from Northwest part of Tigray alone, we brought 17 cases from their house”. In August 2022, he added, “I was with one local organization in Tigray Central Zone called Edaga Arbi, and we had a community awareness creation program on fistula for 30 women attendants; six of them reported that they had fistula. Although three of the six fistula patients had already reported to the nearby health facilities, they could not get treated and they could not come to Mekelle fistula hospital as they cannot afford cost of transportation rehabilitation for longer period up to three months”.

There are several potential risk factors for fistula in the Tigray conflict zone. First, obstructed labor is a common risk factor for fistula. The conflict has resulted in there being limited or an absence of maternal care services, including delivery services. As of May 2022, Tigray Regional Health Bureau [10] revealed that 36/40 hospitals had been fully destroyed or partially damaged, 208/232 health centers had been fully destroyed or partially damaged, and 670/741 health posts had been fully destroyed or partially damaged; moreover 274/308 ambulances had been burned or looted. Such targeted attacks on the health infrastructure have severely affected the maternal health services [4,6,10]. For example, antenatal care (ANC) decreased from 94% in 2019 to 16% in 2022 and skilled delivery from 81% to 21%, leading to obstructed labor during childbirth [10]. A previous study in Adigrat, East Tigray [11], showed that mothers who did not attend ANC were much more likely to develop fistula than those who did, with it being reported that 92.4% of patients with obstetric fistula had no ANC. Worse, there has been an imposed medical siege in the region since June 2021 and the referral system is broken and communication interrupted.

Second, malnutrition is a risk factor for birth difficulties and, subsequently, fistula [12]. Malnutrition causes stunted growth, which leads to an underdeveloped pelvis, and babies may become stuck in the pelvis during delivery, leading to prolonged labor and necrotizing tissues. The United Nations (UN) [13] reported that, in October 2021, 79% of pregnant and lactating mothers were diagnosed with acute malnutrition in Tigray, and over 90% of the population lack regular access to food [14]. Such a high proportion of food-insecure people, resulting from the *de facto* blockade, as it has been described by the UN, has potentially changed the demography of fistula. The United Nations Population Fund (UNFP) [7] revealed that the incidence of fistula in the Tigray conflict zone is now equally high in women from urban and educated backgrounds as it was in the rural, uneducated, and poor women before the start of the conflict. The denial of access to banking services in the region and the lack of a salary for employees for more than 14 months has exacerbated the poverty and lack of finance, which have proved to be further risk factors for fistula.

Third, conflict-related sexual violence commonly leads to trauma and early or unwanted pregnancy, consequently increasing the risk of fistula [8]. Several reports, including Amnesty International reports, have documented the weaponization of sexual violence in Tigray [15]. Taking advantage of impunity, gang-rape in Tigray is characterized by burning the genitals with hot metal or damaging other reproductive or vital organs and insertion of foreign materials into women’s privates [16,17]. Systemic gender-based violence as a cause of fistula, with perpetrators taking advantage of impunity, has also been observed in other conflict zones. For example, 63.4% of the 702 fistula cases that were traumatic in Eastern Congo resulted from sexual violence or gang-rape, whereas 36.6% had obstetric origins [18].

Fourth, sexually transmitted infections (STIs) [19] and tuberculosis [20] are also factors. A significant number of women and girls have been deliberately infected by STIs, including human immune-deficiency virus (HIV) [17], in Tigray. According to Tigray Regional Health Bureau, of those who reported that they were raped, 7.5% were diagnosed with an STI, of which 5% were HIV-positive [21]. Moreover, the fact that 90% of tuberculosis cases in Tigray have been lost to the conflict may lead to an increase in extra-pulmonary tuberculosis, including perianal tuberculosis, a rare cause of fistula. It is known that *Mycobacterium tuberculosis* can also be present in the perianal areas and cause perianal abscesses, leading to abnormal connection of the skin and rectum; i.e., anal fistula. The absence of medication, high rates of malnutrition, and increases in and lack of treatment of other chronic illnesses are exacerbating the immunosuppression of people with tuberculosis, subsequently leading to drug resistance. 

## 2. Conclusions

In Tigray, it is a sad reality that the risk factors for fistula are increasing alarmingly but the services to prevent or cure it are dwindling dramatically or else completely absent. Thus, a coordinated, concerted international effort is needed to: (a) secure peace and stability and lift the siege imposed since the start of the war; (b) increase human and financial resources to support infrastructure, the judicial system, civil society, governance, the fistula clinic in particular, and the entire health system in general; (c) address the question of the accountability and impunity of patrons and perpetrators; (d) ensure health-system support for skilled clinical and other personnel, along with the required infrastructure, equipment, supplies, and logistical resources; (e) strengthen counseling and psychosocial support for treatment and rehabilitation; (f) address special sensitivity and patient–provider interaction issues; (g) collect and handle potential evidence; (h) fund HIV testing services, prophylaxes, etc.; and (i) conduct clinical and operational research to develop best practices and approaches.

## Data Availability

Not applicable.

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
