# Peer review of "Fistula in War-Torn Tigray: A Call to Action"

_ijerph, 2022, doi:10.3390/ijerph192315954_

Round 1
Reviewer 1 Report
An interesting and concise commentary article. The authors make the international plight clear and present some worrying statistics to support their claims and reasoning. I include only some minor remarks for the authors to consider to improve the English.
Minor remarks:
p1, line 30: “highest suffer” should read “highest price”.
p1, lines 36-37: Would read better as “…and sadly, the women living in war-torn Tigray are susceptible to almost all the risk factors.”
p2, line 57: “attack” should be “attacks”
p2, line 65: This needs some rewording. It is not the correlation that is a risk factor, but the malnutrition, etc.
p2, line 68: The United Nations… Same with line 72.
p2, line 70: What does food insecure mean? Consider rewording.
p2, line 79: Typo “balder”.
p2, line 83: “private” should be “privates”.
p2, lines 83-84: This sentence has a couple of typos: “… gender based violence on fistula in conflict zones is also …”
p2, line 88: Should read “A significant number of women and girls…”
p2, line 92: Omit “follow following”. Sentence should read “…Tigray were lost to the conflict…”
p2, line 95: “leading to abnormal connection…”
p3, lines 100-101: This statement might not be true outside of Tigray. Suggest clarifying this, e.g at the start of the sentence, “In Tigray, it is a sad reality…”. Also consider rewording the ending: “…dramatically dwindling, or otherwise completely absent.”
p3 line 102-3: Should read “…siege being imposed since the start of the war…”.
Reviewer 2 Report
Thank you for the opportunity to review this commentary, describing the devastating consequences of the current situation in Tigray, focusing on women’s health.
It is a cry for help in a situation that cannot be solved by the local medical personal.
The current situation, the underlying reasons and potential prerequisites to improve the support of affected women and health professionals is laid out. The authors describe the current situation and point out the potential impact of various factors that might be responsible to contribute to the rising amount of fistulas.
Data from international and local health institutions are included to substantiate the statements.
In conclusion, the contribution highlights an important aspect of women´s health in a war shaken country. Even though no scientific comparison of the data before and current can be provided, pointing out the needs and necessities to overcome the situation is worth to be supported.
If available comparing data from years before the conflict to the current situation would substantiate the paper considerably.
Also: considering the high rate of FGM (Female Genital Mutilation) and early marriage in the country and its potential impact on obstructed labour – commenting on this and correlating it to former data might be interesting.
Some minor comments are mentioned below.
Line 14 … difficulty, trauma and sexually …
> Pregnancy is already mentioned
Linie 31 maybe add reason why the women are rejected. Among others certainly social isolation/stigmatization due to fecal/urinary incontinence.
Line 37 .. tragically happening to the women living in 36 the war-torn Tigray. Fistula cases to have soared as a result of the war
> Is there any data on previous rates? Maybe from the Mekele´s Fistula Clinic?
Line 39 a rare successful
Lines 53, 67 use similar: labour / labor
Line 60 delete “a”
Line 65 malnutrition-stunted growth and birth
Line 66 any relation to early marriage known? Former data available?
Line 71 potentially changing
Line 72 that the incidence of fistula in the conflict zone…
Line 74 educated background, in contrast to rural, uneducated and poor setting before the start of the conflict.
Line 76 and lack of finance, which proved to be another risk factor.
Line 77 Third, conflict related sexual violence is commonly leading to tissue trauma and early or unwanted pregnancy, consequently increasing the risk for fistulas [8].
Line 83-86 revise sentences
Line 88 A significant amount of women….
Line 91 which 5% were HIV..
Line 92 lost to follow-up during the ..
Line 100 dwingling, if not missing at all
Line 103 revise
